# Eye morphogenesis driven by epithelial flow into the optic cup facilitated by modulation of bone morphogenetic protein

Stephan Heermann[1,2]*[†], Lucas Schütz[1], Steffen Lemke[1], Kerstin Krieglstein[2], Joachim Wittbrodt[1]*

[1]Centre for Organismal Studies Heidelberg, Ruprecht Karls Universität, Heidelberg, Germany; [2]Department of Molecular Embryology, Institute of Anatomy and Cell Biology, University Freiburg, Freiburg, Germany

**Abstract** The hemispheric, bi-layered optic cup forms from an oval optic vesicle during early vertebrate eye development through major morphological transformations. The overall basal surface, facing the developing lens, is increasing, while, at the same time, the space basally occupied by individual cells is decreasing. This cannot be explained by the classical view of eye development. Using zebrafish (*Danio rerio*) as a model, we show that the lens-averted epithelium functions as a reservoir that contributes to the growing neuroretina through epithelial flow around the distal rims of the optic cup. We propose that this flow couples morphogenesis and retinal determination. Our 4D data indicate that future stem cells flow from their origin in the lens-averted domain of the optic vesicle to their destination in the ciliary marginal zone. BMP-mediated inhibition of the flow results in ectopic neuroretina in the RPE domain. Ultimately the ventral fissure fails to close resulting in coloboma.

*For correspondence: stephan. heermann@cos.uni-heidelberg. de (SH); jochen.wittbrodt@cos. uni-heidelberg.de (JW)

Present address: [†]Anatomie und Zellbiologie, Heidelberg University, Heidelberg

Competing interests: The authors declare that no competing interests exist.

## Main Text

The bi-layered optic vesicles of vertebrates are formed through a bilateral evagination of the late prosencephalon. In teleosts, this process is driven by a migration of single cells that undergo a subsequent intercalation into the epithelium of the expanding optic vesicle (*Rembold et al., 2006*, *England et al., 2006*, *Sinn and Wittbrodt, 2013*, *Ivanovitch et al., 2013*). The oval optic vesicle develops into a hemispheric bi-layered optic cup through a process that involves major morphological transformations. A long-held view of this process proposes that the lens-averted epithelium of the optic vesicle differentiates into the retinal pigmented epithelium (RPE), while the epithelium facing the lens gives rise to the neuroretina, which subsequently bends around the developing lens (*Chow and Lang, 2001*; *Fuhrmann, 2010*; *Walls, 1942*). This neuroepithelial bending is driven by a basal constriction of lens-facing retinal progenitor cells (RPC) (*Martinez-Morales et al., 2009*) (*Bogdanović et al., 2012*), which ultimately reduces the space occupied by an individual RPC at the basal surface. However, we observed that this is accompanied by a 4.7-fold increase in the overall basal optic cup surface area (*Figure 1A–C*). To identify the cellular origin of this massive increase, we performed in vivo time-lapse microscopy in zebrafish at the corresponding stages, starting at 16.5 hpf (*Figure 1D–L*, *Video 1*), in a transgenic line expressing a membrane-coupled GFP in retinal stem and progenitor cells (Rx2::GFPcaax).

**eLife digest** The eye is our most important organ for sensing and recognizing our environment. In humans and other vertebrates, the eye forms from an outgrowth of the brain as the embryo develops. This outgrowth is called the optic vesicle and it is rapidly transformed into a cup-shaped structure known as the optic cup. Defects in this process prevent the optic cup from closing completely, which leads to a severe condition called Coloboma—one of the most frequent causes of blindness in children.

The optic cup has two distinct layers: the inside layer—known as the neuroretina—contains light sensitive cells and is surrounded by the other layer called the pigmented epithelium. It is thought that the neural retina is made from cells from the side of the optic vesicle that faces the lens, and the pigmented epithelium is formed by cells from the other side of the vesicle. This is a plausible explanation and is well accepted, but it cannot explain how the neuroretina can become five times larger as the cup forms.

Heermann et al. addressed this problem by using four-dimensional in vivo microscopy to follow individual cells as the optic cup forms in living zebrafish embryos. The experiments show that the neuroretina is made of cells from both sides of the optic vesicle. Cells from the back of the optic vesicle (furthest away from the lens) join the rest of the cells by moving around the outside rim of the cup.

Further experiments found that a signaling molecule called BMP—which is crucial to the normal development of the eye—controls the flow of cells around the developing optic cup. This factor needs to be carefully controlled during the development of the eye; when BMP activity was artificially increased, the flow of cells stopped, resulting in neuroretinal tissue developing in the wrong place (in the outer layer of the optic cup).

The experiments also reveal that the stem cells in the retina—which divide to produce new cells throughout the life of the zebrafish—originate from two distinct areas in the optic vesicle.

Heermann et al.'s findings challenge the textbook model of eye development by revealing that cells from both sides of the optic vesicle contribute to the neuroretina and that retinal stem cells originate from a specific place in the developing eye. A future challenge will be to understand how the movement of the cells into the neuroretina is coordinated to make a perfectly shaped eye.

Strikingly, and in contrast to the former model (*Chow and Lang, 2001*; *Fuhrmann, 2010*; *Walls, 1942*), our analysis shows that almost the entire bi-layered optic vesicle gives rise to the neural retina (*Figure 1D–I*), with the marked exception of a small lens-averted patch (see below). The majority of the lens-averted epithelium (*Figure 1D,G*, between arrowheads) serves as a neuro-epithelial reservoir, which eventually is fully integrated into the lens-facing neuro-epithelium (*Video 1*). This occurs through a sheet-like flow of lens-averted cells into the forming optic cup (*Figure 1E,H*). This epithelial flow is independent of cell proliferation (*Figure 2—figure supplement 1*, *Video 2*) as demonstrated by aphidicolin treatment. The process is highly reminiscent of gastrulation movements and explains the marked increase of the lens-facing basal neuroretinal surface area. Notably, a small domain of the lens-averted epithelium exhibits a different morphology and behavior. As optic cup formation proceeds, this region flattens, enlarges, exhibits the morphological characteristics of RPE, and eventually ceases expressing RX2, a marker for retinal stem and early progenitor cells (*Figure 1H*, asterisks, *Video 3*, in between arrows).

Our data highlight that almost the entire optic vesicle contributes to the formation of the neural retina. This new perspective on optic cup formation raises the question of how the elongated oval optic vesicle is transformed into the hemispheric optic cup. We addressed this by 4D imaging of optic cup formation using a nuclear label (H2BGFP) (*Figure 2A*). We found, concomitant with lens formation, a prominent epithelial flow around the temporal perimeter of the forming optic cup. An involution of cells from the domain of the retinal pigmented epithelium (RPE) into the domain of the neuroretina had been proposed (*Li et al., 2000*). Such reorganization of the lens-averted and the lens-facing epithelia, affecting the temporal optic cup, has been subsequently described (*Picker et al., 2009*) and confirmed (*Kwan et al., 2012*). It was proposed that such 'rim movements' could occur around most of the optic vesicle circumference (*Kwan et al., 2012*).

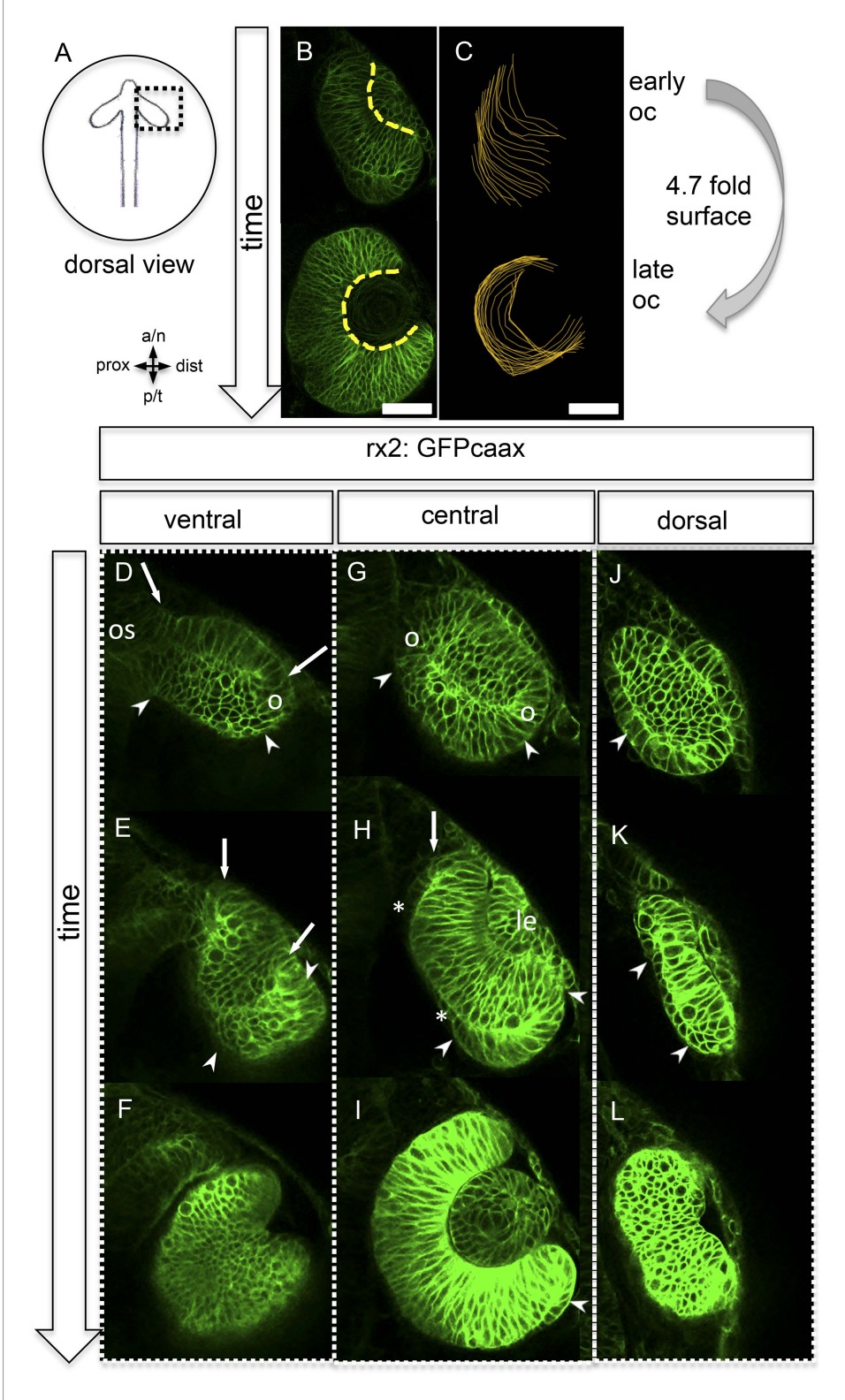

**Figure 1**. Neuroretinal surface increases during optic cup formation by epithelial flow. (**A**) Scheme showing the orientation of the pictures presented in **B**–**L**. (**B**) Basal neuroretinal surface increases from early to late optic cup stage (dashed yellow lines). (**C**) Basal neuroretinal surface was measured in 3D (superimposed orange lines), although RPCs undergo basal constriction during optic cup formation, the surface increases 4.7 fold from early to

*Figure 1. continued on next page*

*Figure 1. Continued*

late optic cup stage, (**D–L**) transition from optic vesicle to optic cup over time, shown at a ventral (**D–F**), a central (**G–I**), and a dorsal (**J–L**) level. The membrane localized GFP is driven by an rx2 promoter (rx2::GFPcaax), which is active in RPCs. The optic vesicle is bi-layered (**D**, **G**, **J**) with a prospective lens-facing (arrows in **D** and **E**) and a prospective lens-averted (arrowheads in **D**, **G**, **J**) epithelium, connected to the forebrain by the optic stalk (os in **D**), at a ventral level both are connected at the distal site (circle in **D**), at a central level both are connected distally and proximally (circles in **G**), notably the morphology of the lens-averted epithelium at a dorsal level is different from central and ventral levels (arrowhead in **J**). Over time at ventral and central levels (**D–F** and **G–I**, respectively), the lens-averted epithelium is being integrated into the forming optic cup (arrowheads in **D**, **E**, **G**, **H**, **I** and arrow in **H**). A patch of cells in the lens-averted domain gives rise to the RPE (asterisks in **H** and arrowhead in **J**, **K**), le: developing lens, os: optic stalk, **B** and **D–L** were derived from 4D imaging data starting at 16.5 hpf (**D**, **G**, **J**), one focal plane is presented as *Video 1*, scalebar in **B** and **C** = 50 µm.

Our data confirm a flow around the temporal perimeter and additionally demonstrate epithelial flow around the nasal perimeter into the forming optic cup. We uncover that the direction of the epithelial flow primarily establishes two distinct neuroretinal domains (nasal and temporal) separated by the static dorsal and ventral poles of the forming eye (*Figure 2D*, *Figure 3A*). We use these poles as dorsal and ventral reference points throughout the manuscript. Importantly, the prominent rotation of the eye cup only occurs after the epithelial flow has ceased (24–36 hpf, *Schmitt and Dowling, 1994*).

The prospective RPE remains in the lens-averted domain and expands in conjunction with the bifurcated flow of the neuroretina from the lens-averted into the lens-facing domain (*Figure 2A,B*, *Video 3*). To further address the transformation of the elongated, oval optic vesicle into the hemispheric optic cup, we quantified cellular movements along the dorso-ventral axis. We found that the most prominent movements leading to the extension in the dorsal ventral

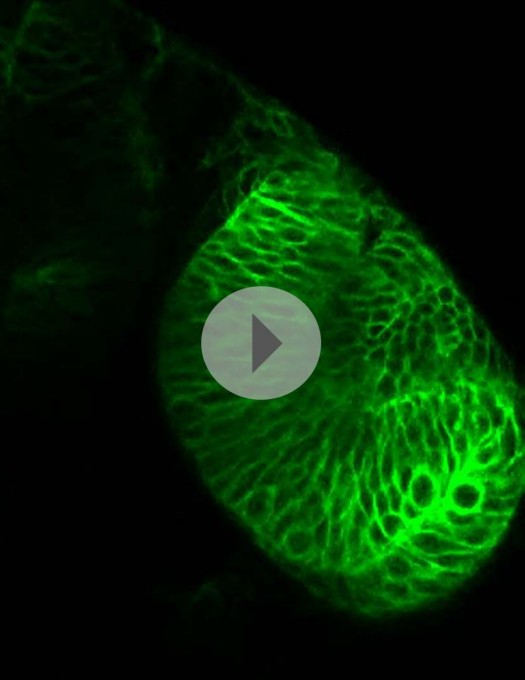

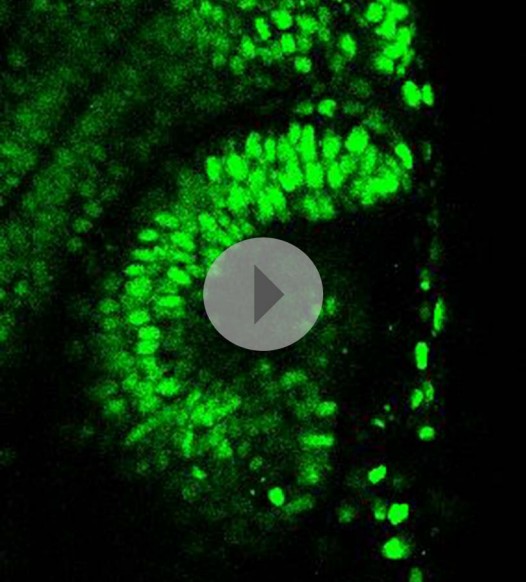

**Video 1.** (related to *Figure 1*) (control) Optic vesicle to optic cup transition visualized with rx2::GFPcaax (orientation as in *Figure 1*) (imaging starts at 16.5 hpf, framerate 1/15 min).

**Video 2.** (related to *Figure 1* and *Figure 2—figure supplement 1*) Aphidicolin treated embryo. (imaging started at 17 hpf, framerate 1/10 min).

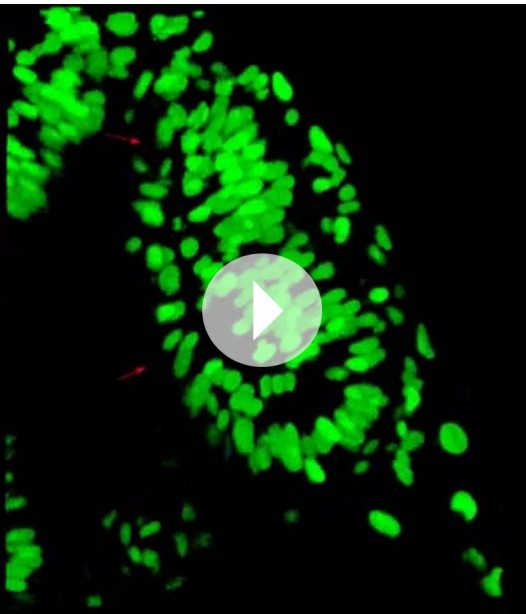

**Video 3.** (related to *Figure 1*) (control) Optic vesicle to optic cup transition visualized by H2BGFP RNA into rx2:: GFPcaax (orientation as in *Figure 2*), data derived from same imaging data as *Video 4*, 3D rendered. Arrows mark the border between future RPE and Neuroretina (imaging starts at 16.5 hpf, framerate 1/10 min).

axis occurred in the ventral domain (*Figure 2C*). A key step in the formation of the ventral neuroretina is the formation of the optic fissure at the ventral pole of the optic vesicle. Lens-averted epithelium flows through this fissure into the forming optic cup to constitute the ventral neuroepithelium (*Figure 2D*). Taken together, we present a model of optic cup formation, driven by gastrulation-like epithelial flow from the lens-averted into the lens-facing epithelium of the forming optic cup. The epithelium flows in two domains around the temporal and nasal rim, respectively and through the optic fissure of the forming optic cup. Overall, this has far-reaching implications for different aspects of eye development. One is the establishment of the retinal stem cell niche in the ciliary marginal zone (CMZ) (*Centanin et al., 2011*), the distal rim of the optic cup/retina.

To address whether the CMZ domain originates from a mixed population of progenitor cells that have been 'set aside', or from a predefined coherent domain, we analyzed the transition from optic vesicle to optic cup in 3D over time (4D) (*Video 4*). By tracking individual cells, we identified the origin of the distal retinal domain, the future CMZ, as two distinct domains (nasal and temporal) within the lens-averted epithelium at the optic vesicle stage (*Figure 3A,B*, *Video 5*). Based on tracking information, we noticed distinct phases during the flow from the lens-averted domain towards the CMZ (*Figure 3D*). Although cells show high motility in an early phase (*Figure 3D*, 1), the directed flow is established only in a later phase (*Figure 3D*, 2), in which cells ultimately flow to the rim of the forming optic cup (*Figure 3D*).

As indicated above, the dorsal pole of the optic vesicle remains static (*Figure 2D*). Thus, the presumptive dorsal CMZ domain either originates from the lens-facing neuroretina or, alternatively, is established secondarily at a later time point, like the ventral CMZ in the region of the optic fissure. The identification of lens-averted domains as the source of the future nasal and temporal CMZ is consistent with the hypothesis of a distinct origin of retinal stem cells. Our data support a scenario in which the entire optic vesicle is initially composed of stem cells that at the lens-facing side respond to a signal to take a progenitor fate.

We propose a tight coupling of morphogenesis with cell determination by inductive signals derived from the surface ectoderm to explain the successive spreading of retinal differentiation from the center to the periphery (*Sinn and Wittbrodt, 2013*). Accordingly, lens-averted stem cells might retain their stem cell fate because they are exposed to that signal at the latest point in time. An alternative hypothesis is that stemness might require an active process at the interface to the RPE; it is also possible that both scenarios are involved. Both scenarios are consistent with the expression pattern of *rx2*, which is initially found in the entire optic vesicle and subsequently is confined to the CMZ. Strikingly, rx2-positive cells of the CMZ represent multipotent retinal stem cells (Reinhardt, Centanin et al., submitted).

We demonstrated that cell motility and thus tissue fluidity are a prerequisite for neuroretinal flow. These characteristics are likely maintained through signaling, raising the question of which system might be involved. A likely candidate might be BMP, which has been linked to mobility in other tissues during development. In heart jogging, for example, BMP has an 'antimotogenic' effect (*Veerkamp et al., 2013*). BMP signaling is important for various aspects of vertebrate eye development such as the enhancement of RPE and the inhibition of optic cup/neuroretina development (*Fuhrmann et al., 2000*; *Hyer et al., 2003*; *Müller et al., 2007*; *Steinfeld et al., 2013*), the formation of the dorso-

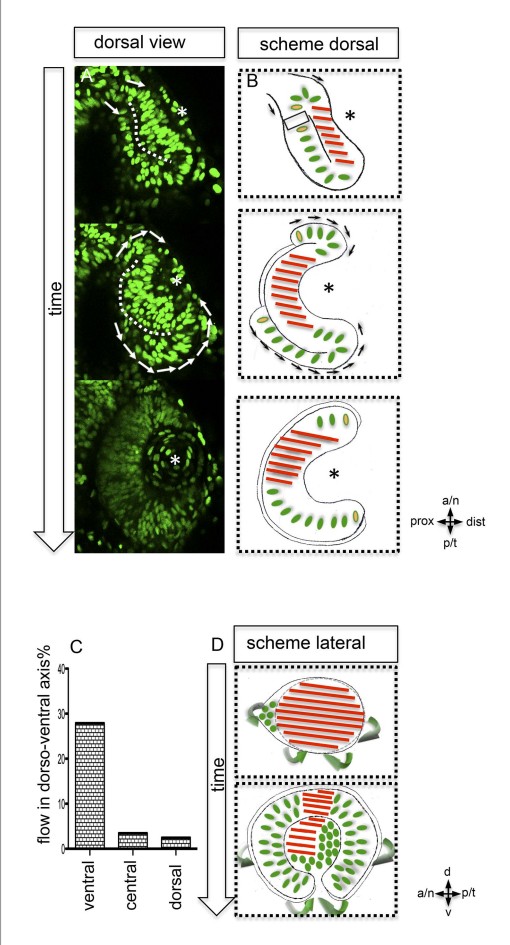

**Figure 2**. Neuroepithelial flow drives morphological changes from optic vesicle to optic cup: the role of the optic fissure and the impact on the forming stem cell domain. (**A**) Dorsal view on optic cup development over time visualized by mosaic nuclear GFP (H2BGFP) (data are derived from 4D imaging data started at 16.5 hpf, one optical section is provided as *Video 4*), while the lens-facing neuroepithelium is starting to engulf the developing lens (asterisk), the lens-averted epithelium is largely integrated into the lens-facing epithelium by flowing around the distal nasal and temporal rims (arrows). A white dotted line marks the border between lens-facing and lens-averted epithelium. (**B**) Scheme showing the key findings of **A**, the lens (asterisk) facing epithelium is marked with red bars. The lens-averted epithelium, which over time is integrated into the lens-facing epithelium is marked with green dots (except the cells at the edges are additionally marked with a yellow core). In between the last cells, which are integrated into the optic cup, the RPE will form in the lens averted domain. (**C**) shows the percentage of movements with a considerable share in dorso-ventral direction for the dorsal, central, and ventral area of the developing eye. In the ventral area of the eye, there is significantly more movement in the dorso-ventral axis, than in the central or dorsal area. (**D**) scheme demonstrating the optic

*Figure 2. continued on next page*

ventral axis (*Behesti et al., 2006*; *Holly et al., 2014*; *Koshiba-Takeuchi et al., 2000*; *Sasagawa et al., 2002*), and the induction of the optic fissure (*Morcillo et al., 2006*). Specific regions of the eye also seem to depend on the modulation of BMP signaling by the expression of a BMP antagonist (*Sakuta et al., 2001*, *French et al., 2009*).

We analyzed BMP signaling activity by assays based on the phosphorylation of the Smads 1/5/8 and the activation of a BMP signaling reporter (*Laux et al., 2011*). BMP signaling is mainly elevated in the temporal domain and to a lesser degree in the nasal domain of the optic vesicle (16.5hpf, *Figure 4A,B,D,E*). At 21.5 hpf BMP signaling is confined to the dorsal pole of the optic cup (*Figure 4C,F*). The transcriptional BMP sensor is activated with a delay and shows a more confined area of activity (compare *Figure 4A–C* to *Figure 4D–F*).

To address the means by which BMP activity is restricted, we analyzed the activity of prominent BMP antagonists follistatin a (fsta) (*Thompson et al., 2005*), and bambi (bambia) (*French et al., 2009*). Fsta was expressed in two domains, a nasal and a temporal domain (*Figure 4G–I* and *Figure 5A*), whereas bambi was only expressed in the temporal domain of the optic vesicle (*Figure 4J,K*) and the dorsal domain of the optic cup (*Figure 4L*). The regions of fsta expression correspond to the domains showing neuroretinal flow during optic cup formation.

To address the importance of localized BMP signaling in wild-type embryos, we expressed BMP4 in the entire eye using an *Rx2* proximal cis regulatory element (*Figure 5B*), which overrides the localized BMP antagonist in the optic vesicle and optic cup.

In BMP reporter fish (*Laux et al., 2011*), we addressed BMP signaling activity under control and experimental conditions. At the optic cup stage, moderate BMP signaling activity was observed in the dorsal retina of control fish (*Figures 4F and 5C*). The pan-ocular expression of BMP4 resulted in a strong response of the reporter, indicating pan-ocular BMP4 signaling (*Figure 5D*).

Strikingly resembling the BMP dependent 'antimotogenic' effect (*Veerkamp et al., 2013*), pan-ocular BMP expression arrested epithelial flow during optic cup formation. Time-lapse in vivo microscopy revealed that cells in the lens-averted part of the future neuroretina remained in the prospective RPE domain and did not contribute to the optic cup (*Video 6 and 7*). This ultimately resulted in an apparently ectopic

*Figure 2. Continued*

vesicle to optic cup transition (lateral view). Notably, the morphological change from the elongated oval optic vesicle to the hemispheric optic cup is driven mainly by the ventral regions (arrows mark the orientation of epithelial flow) (**C** and **D**).

The following figure supplement is available for figure 2:

**Figure supplement 1**. Epithelial flow is independent of cell division.

domain of neuroretina that arose from a morphogenetic failure, rather than from a trans-differentiation of RPE (*Figure 6D*, *Fig. 6—figure supplement 1,2,3*, *Videos 8,9,10*). The severity of the phenotype correlated well with levels of fsta expression in the optic vesicle and was most prominent in the temporal domain of the optic vesicle. These findings highlight the importance of the modulation of BMP signaling for epithelial fluidity during the transformation from optic vesicle to optic cup. We propose that the repression of BMP signaling is crucial to mobilize the lens-averted retinal epithelium, causing it to flow and eventually constitute the neural retina to a large extent.

We further investigated the implications of impaired epithelial flow for subsequent steps of eye development (e.g., fate of the optic fissure). After initiation of neuroretinal differentiation in control embryos, the undifferentiated domains are restricted to the un-fused optic fissure margins and the forming CMZ. Both can be visualized by the expression of *Rx2* (*Figure 6A,B*). The impairment of neuroretinal flow, however, resulted in a mis-organization of the optic fissure. Here, the undifferentiated *Rx2*-expressing domain was found at the ultimate tip of the lens-averted neuroretinal domain, which failed to flow into the optic cup and persisted in the prospective RPE (*Figure 6C*). As a result, the temporal optic fissure margin, in particular, failed to extend into the optic fissure (*Figure 6D–E*). This also holds true, but to a lesser extent, to the nasal optic fissure margin (*Figure 6D*). As a result, the two fissure margins cannot converge resulting in a persisting optic fissure, a coloboma. Macroscopically, the pan-ocular expression of BMP4 results in phenotypes including a 'Plattauge' (flat-eye) (*Figure 6G*), in which the ventral part of the eye is strongly affected and a milder phenotype (*Figure 6F*), in which the ventral retina develops, but with a persisting optic fissure.

It was previously shown that exposing the developing eye to high levels of ectopically applied BMP can cause dorsalization, concomitant with a loss of ventral cell identities (*Behesti et al., 2006*). This is likely the cause for coloboma (*Behesti et al., 2006*; *Koshiba-Takeuchi et al., 2000*, *Sasagawa et al., 2002*). Our data based on stable BMP4 expression (rx2::BMP4) in the entire optic vesicle, however, conclusively show that early BMP4 exposure arrests neuroepithelial flow, resulting in a morphologically affected ventral retina. The ventral expression of vax2 in optic cups of rx2::BMP4 embryos indicates the maintenance of ventral retinal fates and argues against early transdifferentiation/dorsalization induced by BMP (*Figure 6—figure supplement 4*). Remarkably, the remaining lens-averted domain of those embryos, which was ectopically localized and was not integrated into the optic cup, eventually differentiated into neuroretina (*Figure 6—figure supplement 1*), as indicated by the expression of vsx1 (*Kimura et al., 2008*; *Shi et al., 2011*; *Vitorino et al., 2009*) and vsx2 (formerly Chx10) (*Vitorino et al., 2009*). Notably, a localization of neuroretina within the RPE domain might be mistaken for an RPE to neuroretina trans-differentiation, as proposed for other phenotypes (*Araki et al., 2002*; *Azuma et al., 2005*; *Sakaguchi et al., 1997*, *Bankhead et al., 2015*).

Even in amniotes, the histological analyses of consecutive stages of optic cup development are best interpreted as epithelial flow that also enlarges the retinal surface. This can even be appreciated during in vitro optic cup formation using mammalian embryonic stem cells (*Eiraku et al., 2011*).

Taken together, our data clearly show that during optic vesicle to optic cup transformation, the lens-averted part of the optic vesicle is largely integrated into the lens-facing optic cup by flowing around the distal rim of the optic cup including the forming optic fissure. Our data have far-reaching implications on the generation of the retinal stem cell niche of teleosts, as the last cells flowing into the optic cup will eventually constitute the CMZ. We identify a part of the lens-averted epithelium as the primary source of the RPE. The arrest of neuroepithelial flow by the 'antimotogenic' effect of BMP (*Veerkamp et al., 2013*) results in coloboma and thus highlights the importance of the flow through the fissure for the establishment of the ventral optic cup.

It is unlikely that the bending of the neuroretina provides the motor for the epithelial flow; in the opo mutant no ectopic neuroretina can be found, indicating that the flow persists, even in the absence

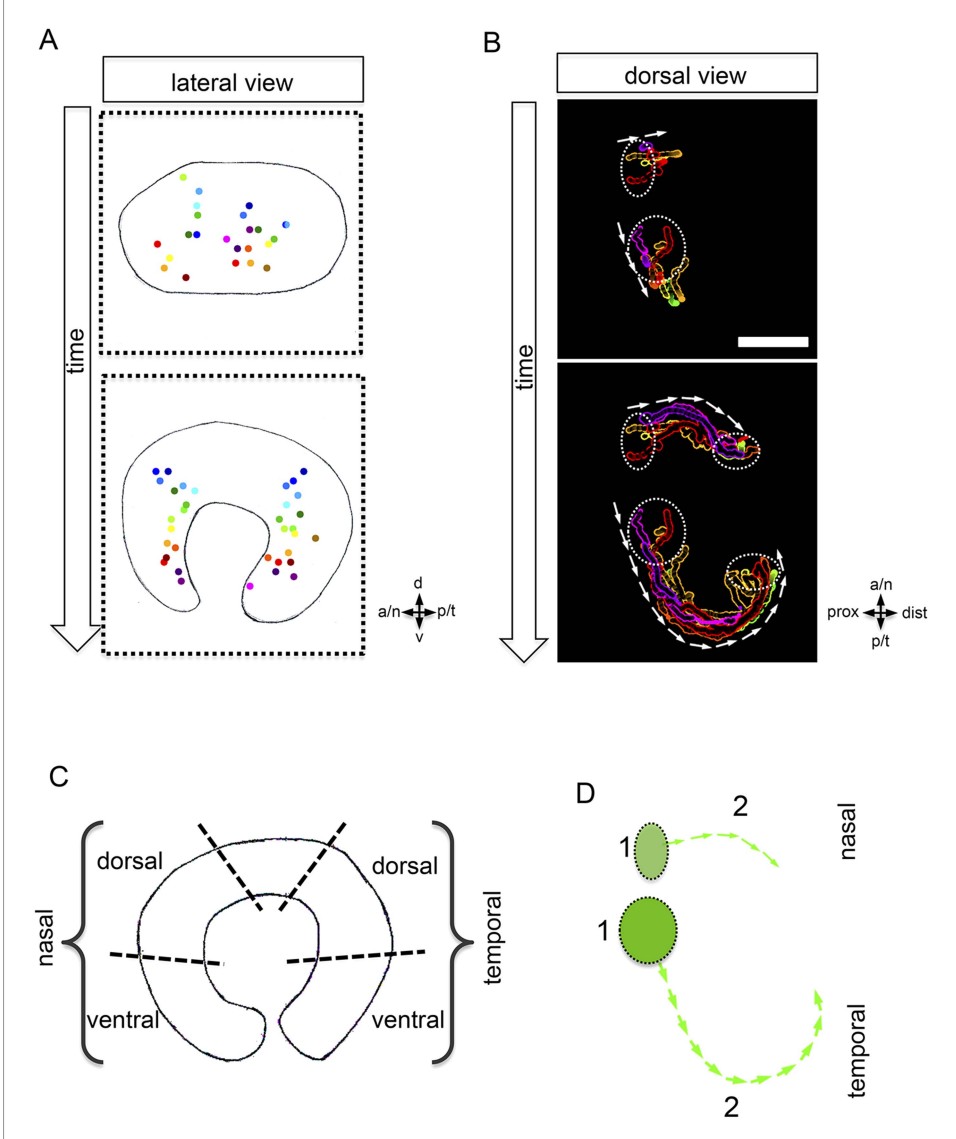

**Figure 3**. Development of the CMZ and quantification of the flow towards this domain. (**A**) scheme of optic cup development (lateral view over time) including the results of nuclear tracking from the presumptive CMZ back in time to the lens-averted epithelium, remarkably two distinct domains became apparent within the lens-averted epithelium as the source for the presumptive CMZ. (**B**) Establishment of the presumptive CMZ domain (dorsal view), nuclear tracking of cells (maximum projection) from the lens-averted domain (encircled in upper picture) eventually residing in the forming CMZ (additionally encircled in lower picture), scalebar = 50 µm. (**C**) Scheme showing the optic cup from the lateral side. For quantification four domains were selected, nasal–dorsal, nasal–ventral, temporal–dorsal, and temporal–ventral. Note that the dorsal distal domain is only assembled secondarily and the ventral pole shows the optic fissure. (**D**) Based on differential effective distance, effective speed, and directionality, the migration distance was divided in two phases in the nasal and temporal domain, respectively.

of optic cup bending (*Bogdanović et al., 2012*). Consequently, forces established outside the neuroretina are likely to drive the flow. One tissue potentially involved is the mono-layered-forming RPE. We speculate that this tissue contributes to the flow by changing its shape from a columnar to a flat epithelium, massively enlarging its surface (*Figure 1J–K*, *Video 3*). This remains an interesting point, in particular given that epithelial flow is maintained even if cell proliferation is inhibited in both neuroretina and RPE.

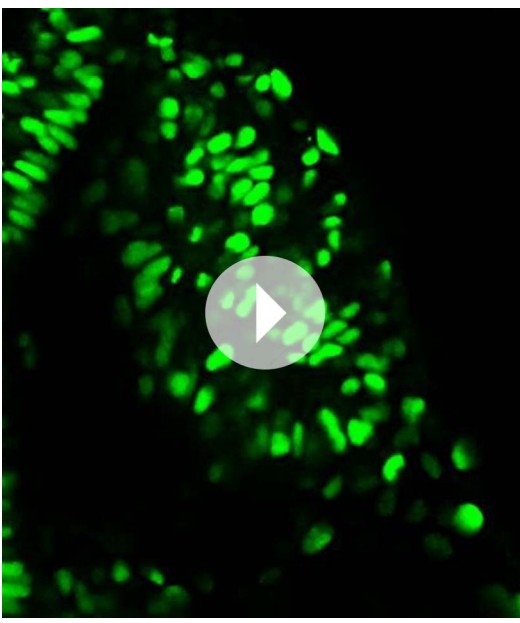

**Video 4.** (related to *Figure 2*) (control) Optic vesicle to optic cup transition visualized by H2BGFP RNA into rx2:: GFPcaax (orientation as in *Figure 2*) (imaging starts at 16.5 hpf, framerate 1/10 min).

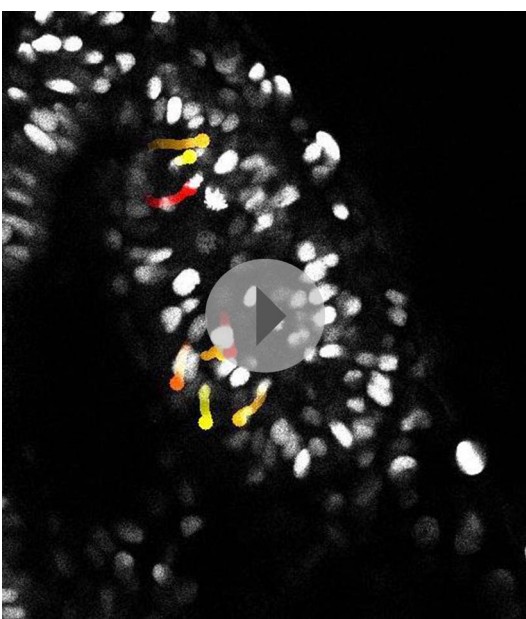

**Video 5.** (related to *Figure 2*) (control) Optic vesicle to optic cup transition visualized by H2BGFP RNA into rx2:: GFPcaax (orientation as in *Figure 2*), data as in *video 4* with tracked cells (maximum projection) to the presumptive CMZ.

## Materials and methods

### Transgenic zebrafish and Injections

BMP4 was cloned via directional Gateway from zebrafish cDNA into a pEntr D-TOPO (Invitrogen, Germany) vector with the following primers: forw: 5′ CACCGTCTAGGGATCCCTTG TTCTTTTTGCAGCCGCCACCATGATTCCTGG TAATCGAATGCTG 3′, rev: 5′ TTAGCGGCA GCCACACCCCTCGACCAC 3′.

The expression construct was assembled via a Gateway reaction using Tol2 destination vector containing a cmlc: GFP (*Kwan et al., 2007*), a 5′ Entry vector containing an rx2 promoter (*Martinez-Morales et al., 2009*), the vector containing the BMP4 and a 3′Entry vector containing a pA sequence (*Kwan et al., 2007*). The construct was co-injected with mRNA encoding Tol2 transposase into the cytoplasm of zebrafish eggs at the one cell stage. Stable lines were preselected based on GFP expression in the heart (cmlc2::GFP), raised and validated in F1 and subsequent generations. Lines were maintained as closed stocks and crossed to other lines as indicated in the manuscript.

The rx2::GFPcaax construct was assembled with the 5′ and 3′ components described above and GFPcaax in the pEntr D-topo vector via Gateway (Invitrogen) and co-injected with mRNA encoding Tol2 transposase into the cytoplasm of zebrafish eggs at the one cell stage. Stable lines were preselected based on GFP expression in the heart (cmlc2::GFP), raised, and validated in F1 and subsequent generations. Lines were maintained as closed stocks and crossed to other lines as indicated in the manuscript.

The BRE::GFP zebrafish line (*Laux et al., 2011*) was kindly provided by Beth Roman. The Vsx1::GFP zebrafish line (*Kimura et al., 2008*; *Shi et al., 2011*; *Vitorino et al., 2009*) was kindly provided by Lucia Poggi. The Vsx2::RFP zebrafish line (*Vitorino et al., 2009*) was kindly provided by the lab of William Harris.

Where indicated RNA for H2BGFP (nuclear localized GFP) (37 ng/µl) was injected into 1–8 cell staged zebrafish embryos enabling 4D imaging of mosaically nuclear labeled zebrafish.

### Drug treatment with aphidicolin

Zebrafish embryos were treated with aphidicolin (10 µg/ml, Serva, Germany) in order to inhibit cell proliferation. 12 embryos were treated with aphidicolin. 4D imaging was performed on one with an aphidicolin pretreatment of 5 hr. The efficacy of the treatment was addressed by analyzing nuclei in mitosis (positive for the expression of phospho-histone H3. At 21.5 hpf pHH3 positive nuclei were counted in central sections of four control

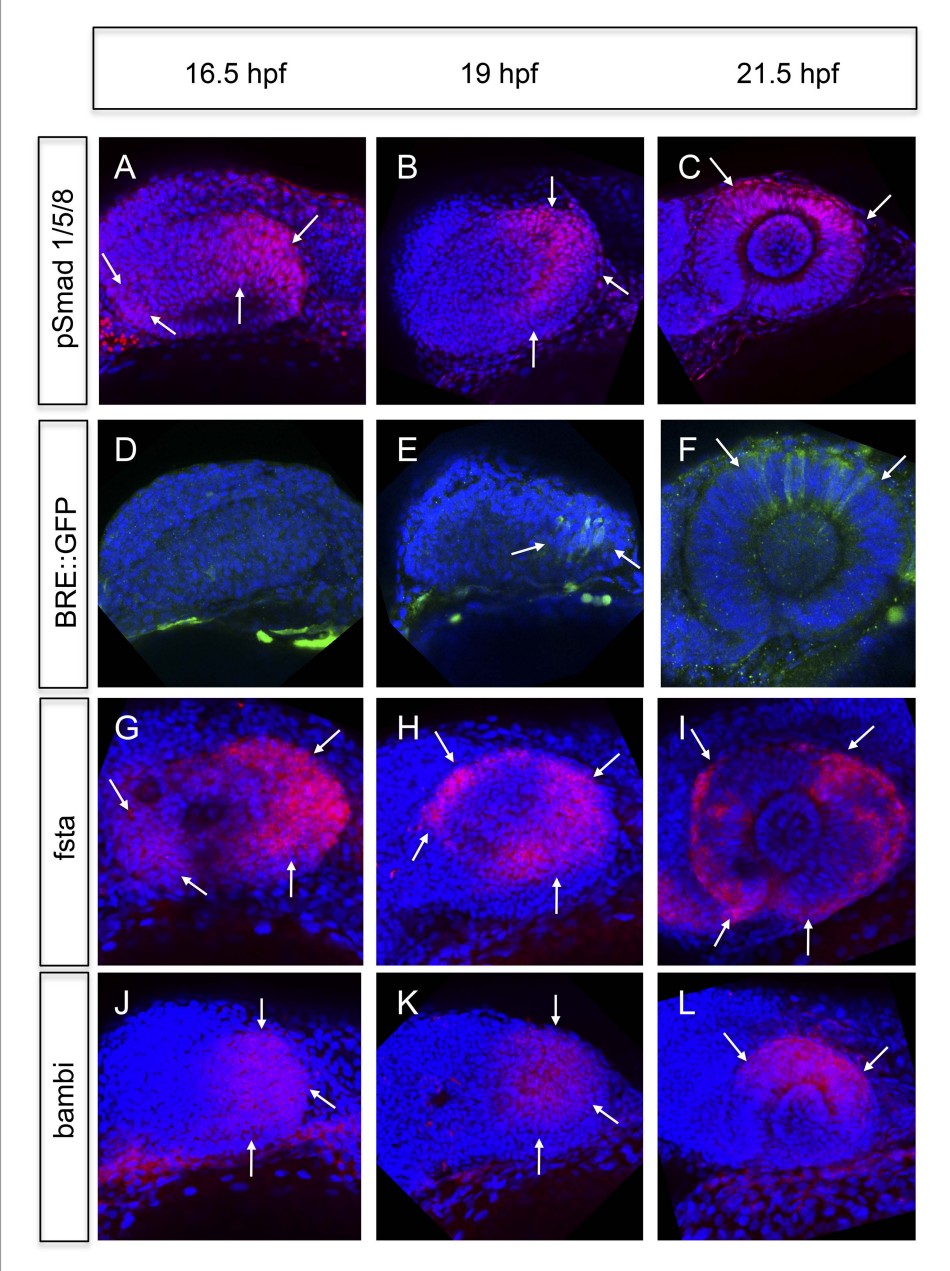

**Figure 4**. Analyses of BMP signaling and expression of BMP antagonists during development at 16.5 hpf, 19 hpf, and 21 hpf embryos are presented in a lateral view nasal left. (**A**–**C**) pSmad 1/5/8 immunohistochemistry (red) and DAPI nuclear staining. Activated BMP signaling can be appreciated mainly in the temporal domain of the optic vesicle (arrows) (**A**–**B**) and in the dorsal domain of the optic cup (arrows) (**C**). At 16.5 hpf, a small domain of activated BMP signaling is visible in the nasal optic vesicle (arrows) (A). (**D**–**F**) Immunohistochemically enhanced BRE::GFP (green) and DAPI nuclear staining. Activated BMP signaling can be appreciated in the temporal late optic vesicle (arrows) (**E**) and the dorsal optic cup (arrows) (**F**). Hardly any activity can be detected in the optic vesicle at 16.5 hpf. Note the delay of activity in comparison to pSmad 1/5/8. (**G**–**I**) Whole mount in situ hybridizations with a fsta probe (Fast Red) and DAPI nuclear staining. In the optic vesicle as well as in the optic cup two domains (nasal and temporal) of fsta expression can be seen (arrows). (**J**–**L**) Whole mount in situ hybridizations with a bambia probe (Fast Red) and DAPI nuclear staining. Bambi expression can be seen in the temporal domain of the optic vesicle (arrows) (**J**–**K**) and in the dorsal domain of the optic cup (arrows) (**L**).

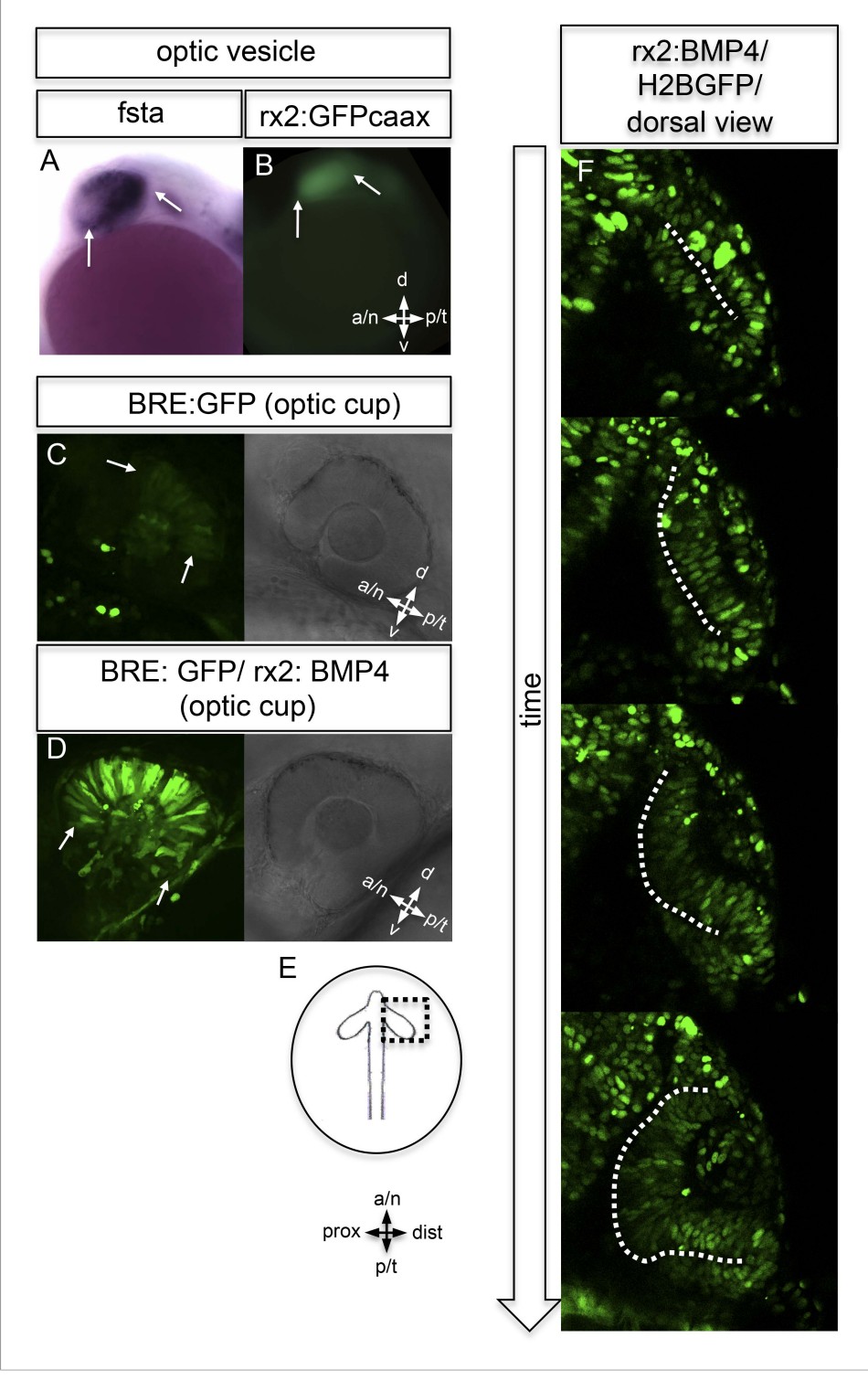

**Figure 5**. BMP antagonism drives neuroepithelial flow during optic cup formation. (**A**) whole mount in situ hybridization for fsta (NBT/BCIP) (17.5 hpf). (**B**) GFP expressed in the optic vesicle (arrows) of an rx2::GFPcaax zebrafish embryo (16.5 hpf), (**C–D**) GFP driven by the BRE and transmission/brightfield image for orientation. Strong GFP expression can be observed in the eye when BMP is driven under rx2 (arrows in **D**), whereas only mild GFP can be observed in controls (arrows in **C**). (**E**) Scheme showing the orientation of the pictures presented in **F**, (**F**) optic cup development over time of an rx2::BMP4 embryo. Cells are visualized by nuclear GFP (H2BGFP). A dotted line is indicating the border between lens-averted and lens-facing epithelium. Remarkably, the pan-ocular driven BMP

*Figure 5. continued on next page*

*Figure 5. Continued*

resulted in persisting lens-averted domains. The data presented in **F** are derived from 4D imaging data (start at 16.6 hpf) one optical section is also presented as *Video 6*.

---

(untreated embryos from the same batch) (average: 21) and experimental (average: 6) retinae, respectively.

## Quantification of optic cup surface

Optic cup surfaces were measured with the help of FIJI (ImageJ NIH software). The mean of the length of the measured lines (*Figure 1C*) of two adjacent optical sections was multiplied by the optical section interval.

## Microscopy

Confocal data of whole mount immunohistochemical stainings a Leica (Germany) SPE microscope was used. Samples were mounted in glass bottom dishes (MaTek, Ashland, MA). Olympus (Germany) stereomircoscope was used for recording brightfield images of rx2::BMP4 hatchlings and the overview of the expression of rx2::GFPcaax. For whole mount in situ data acquisition, a Zeiss (Germany) microscope was used. Time-lapse imaging was performed with a Leica SP5 setup which was upgraded to a multi photon microscope (Mai Tai laser, Spectra Physics, Germany). It was recorded in single photon modus and multi photon modus. For time-lapse imaging, embryos were embedded in 1% low melting agarose and covered with zebrafish medium, including tricaine for anesthesia. Left and right eyes were used and oriented to fit the standard dorsal view or view from the side.

## Whole mount in situ hybridization

Whole mount in situ hybridization was performed with probes for fsta bambia and vax2. The probes were selfmade. Sequences were amplified by PCR from zebrafish cDNA and subcloned into

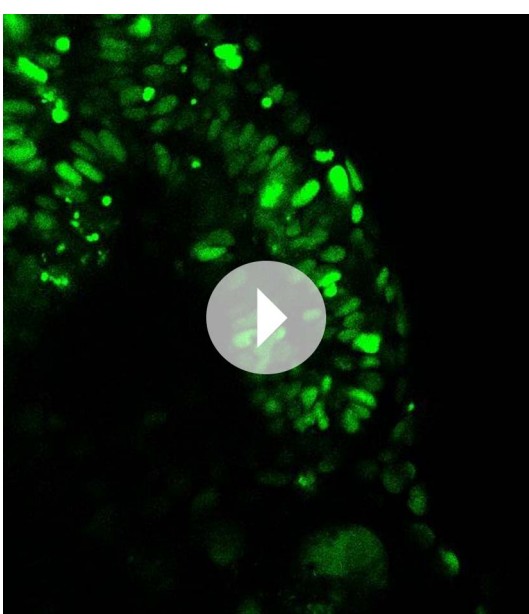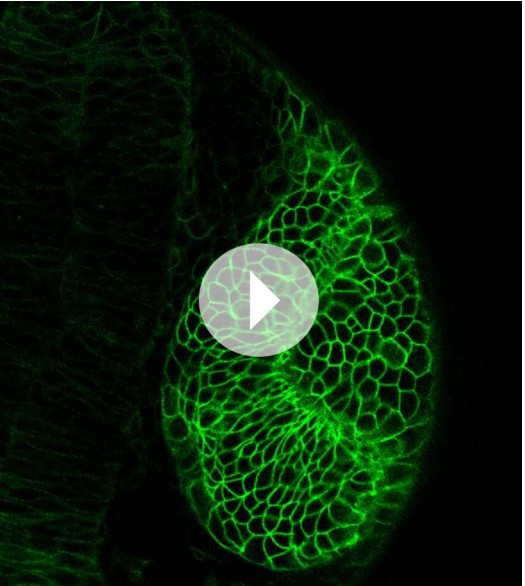

**Video 6.** (related to *Figure 4*) (rx2::BMP4) Optic vesicle to optic cup transition visualized by H2BGFP RNA (orientation as in *Figure 3*) (imaging starts at 16.5 hpf, framerate 1/15 min).

**Video 7.** (related to *Figure 4*) (rx2::BMP4) Optic vesicle to optic cup transition visualized by rx2::GFPcaax (orientation as in *Figure 3*) (imaging starts at 19 hpf, framerate 1/15 min).

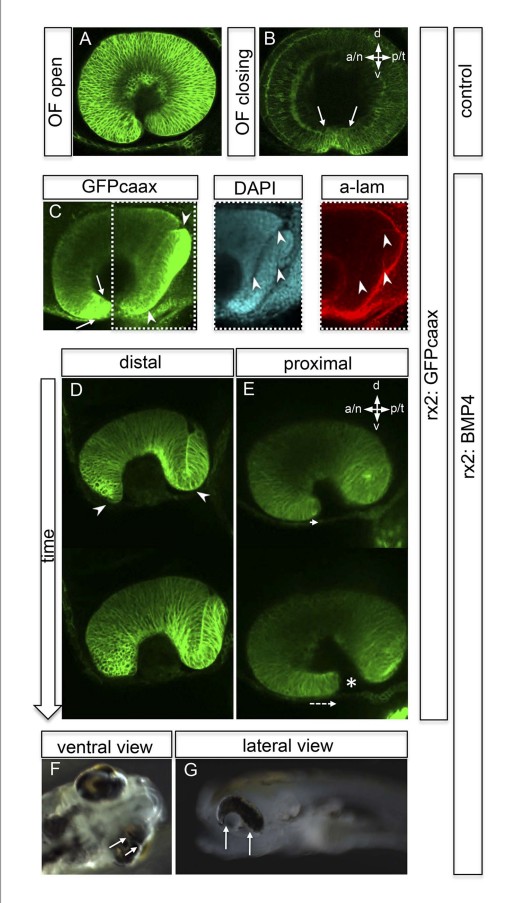

**Figure 6**. Impaired eye gastrulation results in coloboma. (**A–B**) Membrane-localized GFP (rx2::GFPcaax) in a developing eye during optic fissure closure (**A** = early, **B** = late) (lateral view, derived from imaging data, (**A**) start at 24 hpf, (**B**) after 34 hr imaging at 22°C). Rx2 is expressed in retinal stem cells/RPCs (**A**) and after NR differentiation is additionally expressed in photoreceptors and Müller Glia (**B**) while its expression is maintained in retinal stem cells of the CMZ (Reinhardt and Centanin et al., submitted). The optic fissure margins are still undifferentiated (arrows in **B**), (**C**) developing eye of rx2::BMP4 fish (lateral view), membrane-localized GFP (rx2:: GFPcaax, anti-GFP immunointensified), DAPI nuclear labeling and anti-laminin immunostaining, the optic fissure is visible, noteworthy the temporal retina is misshaped and folded into the RPE domain (best visible in DAPI, arrowheads), and located on a basal membrane (arrowheads in anti-laminin), especially the temporal optic fissure margin (arrowheads in GFPcaax) is located in the folded part of the temporal retina and not facing the optic fissure (arrows in GFPcaax) (24 hpf) (**D–E**) impaired optic fissure closure in rx2::BMP4 embryos over time at a proximal (**E**) and a distal (**D**) level. (Data obtained from 4D imaging of rx2 ::BMP4/ rx2::GFPcaax started at 21.5 hr. Data are also presented as **Video 10**.) Importantly, next to the affected temporal optic cup also the nasal optic cup is mis-folded (arrowheads in **D**).
*Figure 6. continued on next page*

pGEMTeasy vector (Promega, Germany). In vitro transcription was performed with Sp6/T7 Polymerase. Hybridization was largely performed according to *Quiring et al. (2004)*. The Probe bas visualized with NBT/BCIP (Roche, Switzerland) or Fast Red (Roche) as indicated.

## Whole mount immunohistochemistry

Immunohistochemistry was performed according to a standard whole mount immunohistochemistry protocol. Briefly, embryos/hatchlings were fixed, washed, bleached (KOH/$H_2O_2$ in PTW), and blocked (BSA [1%], DMSO [1%], Triton X-100 [0.1%], NGS [4%], PBS [1×]). In case of anti-pSmad 1/5/8 immunohistochemistry embryos were additionally treated with proteinase K (10 μg/ml, 16.5 hpf: 5 min, 19 hpf and 21.5 hpf: 6 min). Samples were incubated in primary antibody solution (anti-laminin, 1:50, Abcam, Germany) (anti-GFP 1:200, life technologies, Germany) (anti-dsRED, Clontech, Germany) (anti-pHH3, 1:100, Milipore, Germany) (anti-pSmad1/5/8, 1:25, Cell Signaling, Germany) in blocking solution. Samples were washed and incubated in secondary antibody solution (anti-rabbit Dylight, 1:300, anti-chicken Alexa 488, 1: 300, Jackson, UK) with DAPI (stock: 2 μg/ml, 1: 500) added. Consecutively, samples were washed and mounted for microscopy.

## Quantification of dorso-ventral movement

The amount of movement in the dorso-ventral axis was quantified using a supervoxel based Optical Flow algorithm (*Amat et al., 2013*). The pixel wise output was visualized by applying a spherical coordinate system to the eye using a custom made ImageJ plugin (*Source code 1*: file plugin). The color coding is based on the sign of the polar angle theta and the sign of the azimuth angle phi, as well as on their respective combinations. The quantification was performed by counting the labeled pixels in an ImageJ macro (*Source code 1*: file macro).

## Cell tracking

Cells were tracked manually using MtrackJ (*Meijering et al., 2012*) in Fiji (ImageJ) (*Schindelin et al., 2012*) back in 4D stacks to their original location or until lost. Only tracks with a significant length were used for the visualizations. Centered on the track cells are represented as spheres. Partially results are presented in a side view where the dorso-

*Figure 6. Continued*

Remarkably, however, the nasal optic fissure margin extents into the optic fissure (dashed arrow in **E**) but the temporal optic fissure margin does not, likely being the result of the intense mis-bending of the temporal optic cup. This results in a remaining optic fissure (asterisk in **E**). (**F**–**G**) Brightfield images of variable phenotype intensities observed in rx2::BMP4 hatchlings.

The following figure supplements are available for figure 6:

**Figure supplement 1**. Postembryonic eye development of rx2::BMP4 hatchlings.

**Figure supplement 2**. Lateral view on optic cup development over time, rx2::GFPcaax (control) is compared to rx2::BMP4 at proximal and distal levels.

**Figure supplement 3**. Dorsal view on optic cup development of an rx2::BMP4 embryo over time at ventral vs central/ dorsal levels.

**Figure supplement 4**. Ventral retinal identity remains in rx2::BMP4 embryos.

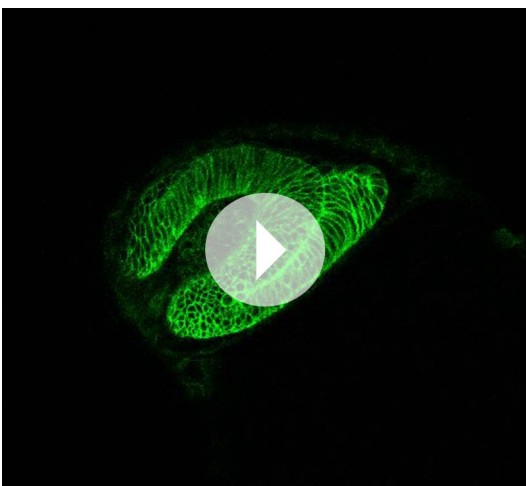

**Video 9.** (rx2::BMP4) Optic cup development recorded with rx2::GFPcaax (lateral view) (imaging starts at 20 hpf, framerate 1/10 min).

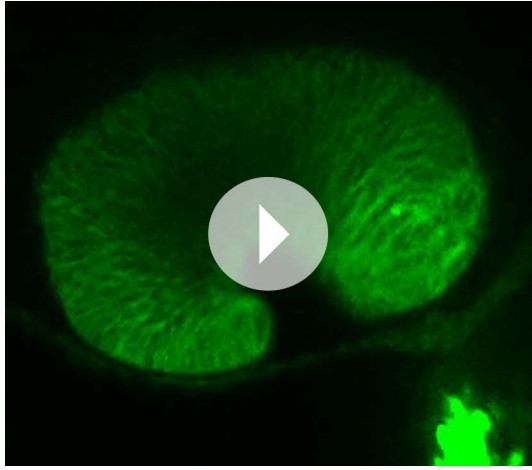

**Video 10.** Proximal domain of an rx2::BMP4 embryo showing an impaired optic fissure closure (orientation as in *Figure 5E*) (imaging starts at 21.5 hpf, framerate 1/10 min).

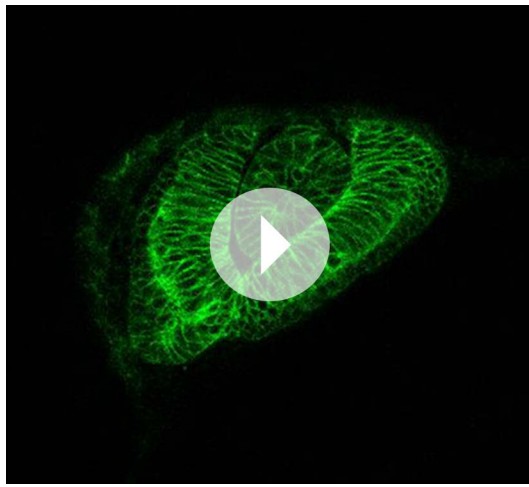

**Video 8.** Control to *video 9*, optic cup development recorded with rx2::GFPcaax (lateral view) (imaging starts at 20 hpf, framerate 1/10 min).

ventral axis originally represented as the z-axis has now become the y-axis. A factor of 10.5703 is introduced in order to adjust the data of the former z-axis to the other two axes. The color coding is done by choosing colors from an 8 bit lookup table and applying them from the dorsal to the ventral side based on the end of the track. Partially tracking results are presented as tailed spheres. The spheres are based on the tracking data using an average over the last three timepoints. The image is stretched in the z-axis using a factor of 10.5703, to adjust the scale to the x and y axes. Tails are created using a lookup table with 16 different shades per color. The respective shade is defined by the distance and difference in time between the recent position and the position on the tail.

## Acknowledgements

We want to thank Karin Schumacher for generously providing space for finishing the work on this manuscript, Lea Schertel for excellent technical assistance, members of the Wittbrodt lab for material (P Haas and A Schmidt for the rx2: GFP construct and zebrafish line, B Höckendorf and M Stemmer for H2BGFP mRNA, B Höckendorf for FIJI related input) and constructive stimulating feedback, B Roman for the BRE: GFP zebrafish, William Harris for the vsx2: RFP zebrafish line and Lucia Poggi for making the vsx1: GFP zebrafish line available. We also want to thank Russ Hodge for valuable input on editing the manuscript. LS is recipient of a fellowship from the Hartmut-Hoffman-Berling International Graduate School (HBIGS). This work was supported by the Deutsche Forschungsgemeinschaft (JW, KK, SL) and the ERC (JW).

## Additional information

### Funding

| Funder | Grant reference number | Author |
|---|---|---|
| Deutsche Forschungsgemeinschaft | DFG/ Molecular mechanisms regulating optic fissure closure | Kerstin Krieglstein, Joachim Wittbrodt |
| European Research Council (ERC) | Advanced Grant (AdG), LS3, ERC-2011-ADG | Joachim Wittbrodt |

The funders had no role in study design, data collection and interpretation, or the decision to submit the work for publication.

### Author contributions

SH, Final approval of the version to be published, Conception and design, Acquisition of data, Analysis and interpretation of data, Drafting or revising the article; LS, Final approval of the version to be published, Acquisition of data, Analysis and interpretation of data, Drafting or revising the article; SL, Final approval of the version to be published, Analysis and interpretation of data, Drafting or revising the article; KK, Final approval of the version to be published, Conception and design, Drafting or revising the article; JW, Final approval of the version to be published, Conception and design, Analysis and interpretation of data, Drafting or revising the article, Contributed unpublished essential data or reagents

## Additional files

### Supplementary file

• Source code 1. Source Code .zip contains: Plugin: imageJ plugin for visualization of dorso-ventral movements (please see 'Materials and methods' section: quantification of dorso-ventral movement). Macro: imageJ macro for counting of labeled pixels (please see 'Materials and methods' section: quantification of dorso-ventral movement).

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
