## [Decision Letter]

Thank you for sending your work entitled “Epithelial flow into the optic cup facilitated by suppression of BMP drives eye morphogenesis” for consideration at *eLife*. Your article has been favorably evaluated by Diethard Tautz (Senior editor) and three reviewers, one of whom, Tanya Whitfield, is a member of our Board of Reviewing Editors.

The Reviewing editor and the other reviewers discussed their comments before we reached this decision, and the Reviewing editor has assembled the following comments to help you prepare a revised submission.

While all three reviewers find the study interesting, there are some substantive reservations over the manuscript in its current form. A major criticism is the lack of citation and acknowledgement of previous literature. A number of papers have been suggested by the reviewers, all of which should be considered, cited and discussed carefully. In addition, the conclusions and interpretation of the findings should be adjusted to take full account of these earlier studies.

In addition, some further experimental work should be done to strengthen the manuscript. A fuller description, at multiple embryonic stages, of the BRE:GFP expression domains in the eye, and correlation with expression of both P–Smads and BMP inhibitors (fsta, bambi), is required. A later stage in situ showing vax2 expression following BMP overexpression should also be shown. If additional data (such as the origin of the stem cell compartment, or tracing the patch of lens-averted epithelium that forms the RPE) can be extracted from the existing movies (or generated with new ones), this information would be extremely valuable and would enhance the significance and novelty of the findings.

A number of other suggestions have been suggested by the reviewers to improve the manuscript, including adding more detail to the Experimental Procedures section and clarifying the data provided in the Table. It will be essential to provide accurate embryo staging and labeling throughout.

The full reviews are appended below.

Reviewer #1:

This is an interesting report documenting the cell movements that contribute to formation of the zebrafish neuroretina. The authors demonstrate that much of the retina derives from lens-averted epithelium, which moves round and into the lens-adjacent layer in directed movements of the cell sheet akin to gastrulation. These movements continue in the absence of cell proliferation, and account for the apparently paradoxical increase in surface area of the neuroretina, despite the decrease in apical surface area of individual cells in this tissue. It is proposed that a small patch of the lens-averted epithelium gives rise to the retinal pigmented epithelium (RPE).

Some of the imaging is spectacular and supports the conclusions of the paper well. However, there are also a number of deficits that should be addressed. These are listed below.

General points:

The authors place great importance on the retinal stem cell niche, but have no markers to show this population. Can they be visualised in some way?

They also place importance on the role of the developing RPE in driving the cell movements, but the cell movements of this layer are not described well at all, and the small patch of cells that is proposed to become the RPE is not identified or followed in any of the figures or movies. It will be important to show this to confirm the assertion that only a small patch of the lens-averted epithelium develops into the RPE.

Numbers and quantitation: more detail is needed throughout, e.g. for the phenotypes of 'variable expressivity' (eleventh paragraph) resulting from pan-ocular expression of BMP4.

Figures:

Scale bars are needed on the figures, especially on Figure 1, where the surface area is estimated from linear dimensions drawn onto sections. The scale is needed to ensure the data shown in the figure panels match up with the values listed in the Table.

Stages/hours post fertilization should also be listed in the legend or directly on the panels for all figures.

Figure 1–figure supplement 1 should show a control (H2BGFP without aphidicolin). If this is provided by Figure 2, there should be a reference to this figure in the legend to Figure 1–figure supplement 1, so that a comparison can be made. Are the small bright spots the dying cells? How many embryos were treated, and with what concentration of aphidicolin? This information does not appear to be provided anywhere. How was the block in cell proliferation confirmed? Was this by counting nuclei, or lack of mitotic figures?

Figure 4–figure supplement 1A, panel 2: This picture is not of publication quality, it is completely out of focus, with no morphological detail visible.

Figure 4–figure supplement 2: It would be helpful to show here a stage series, showing when the BMP reporter normally begins to be expressed in the eye, and how levels of expression compare with other areas of BMP activity in the embryo. Ideally, an additional confirmation of BMP activity, e.g. by P-Smad levels, should be shown; alternatively, more information about this transgenic line or a citation is needed in the Experimental Procedures.

Figure 5–figure supplement 1 merge: The box drawn in the first panel does not appear to correspond to the area shown in the enlargements.

Figure 5–figure supplement 4 needs repeating and improving. A control is needed to show expression levels in non-transgenic embryos. A higher power picture of the staining in the eye would be useful; the outline and arrows currently overlie the staining, making it difficult to see.

Experimental Procedures: More detail is needed for description of the transgenic lines. Are these already published? In which case, a citation is needed. If not published, more description of the constructs used is needed. A ZFIN reference should be given for each line, if available.

Immunohistochemistry: Not enough detail is given here for others to be able to replicate the experiments. What do the acronyms stand for, what dilutions of primary and secondary antibodies were used, and what concentrations of DAPI, DMSO, Triton, etc.?

Table:

The Table is difficult to understand. What are the numbers? Do they refer to measurements from a single embryo, or are they mean values from multiple embryos? (In which case, SEM or SD should be given). If from a single embryo, it is difficult to draw any firm conclusions from the data. In the text, phase 2 is described as a fast, smooth flow, but in all cases the effective speed shown during phase 2 in the table appears to be slower than the values shown for phase 1. The total distance moved in phase 2 needs to correlate with the distances shown on the figures, which is why scale bars on the figures are needed. It would be helpful to have a full stop instead of a comma for the decimal point, as 10,000 could be read as 'ten thousand'.

Movies:

The movies are very helpful, but more information is needed in the legends that describe these. What are the times over which they were taken, what was the frame rate, and what stage are the embryos? As for the figures, this information is needed to correlate with the values shown in the table, and should be given in the movie legends. Video 8 is of lower quality and resolution than the others, and should be improved.

Reviewer #2:

This manuscript analyses the cell dynamics accompanying optic cup formation in the zebrafish and shows that cells in the “lens-averted” region of the optic vesicle are incorporated into the “lens-facing” region. It further shows that the inhibition of BMP is required for this process to occur efficiently. The study has interesting implications to understand the origin of optic cup domains. In addition, it provides a new interpretation for phenotypes that up to now had been interpreted as transdifferentiation of RPE into retina, and highlight the power of live imaging to provide an accurate interpretation of phenotypic outcomes. I consider the work a valuable contribution to our current knowledge on eye morphogenesis and patterning. However, I have important concerns with the way the significance of previous works analyzing this same process has been minimised by the authors, and with the interpretation of some of their observations.

The finding of optic vesicle cells ingressing into the future retinal layer from the prospective RPE layer is not new. It has been extensively described by Picker et al., PLoS Biology, 2009 and Kwan et al., Development, 2012. I disagree with the way the authors seem to remove significance from those papers, and I consider they should be appropriately cited. In particular, Picker et al. clearly and extensively showed that prior to optic cup formation, the optic vesicle is mostly formed by future retinal cells, and that the lens-averted layer of the primordium constitutes the origin of the temporal retinal domain.

This does not mean that I do not consider the current work valuable. The authors do provide a detailed analysis of the consequences this morphogenetic process has to understand the origin of different optic cup domains (in particular the CMZ and the RPE), and dissect part of the molecular mechanisms controlling it.

The authors state that follistatin is expressed in the temporal domain, yet their in-situ (Figure 4) clearly shows expression in the prospective dorsal portion of the optic vesicle. At early stages, the optic vesicle in zebrafish has not acquired its final disposition and temporal is located along the ventral portion of the optic vesicle. Follistatin expression is stronger at the distal-most region of the optic vesicle, both dorsal and ventral, which corresponds to prospective dorsal (see Veien et al., Development, 2008). Indeed, this would be more consistent with previous reports showing a requirement of differential BMP signalling along the dorso-ventral, and not the naso-temporal, axis.

In the same line with my first comment, I find the authors should be more rigorous with their citation of previous work. In the first sentence of their manuscript and based on a previous report from the authors from 2006, they state that optic vesicle evagination occurs by single cell migration. However, since then there have been additional papers analyzing dynamically this process and showing results that support alternative explanations for the active behaviours of eye field cells. These papers (England et al., Development, 2006 and Ivanovitch et al., Dev Cell, 2013) should be acknowledged, and the first statement softened.

Reviewer #3:

Given the involvement of BMP signaling in ocular patterning, it is exciting to find that overexpression of BMP can affect specific cell movements. The manuscript contains some interesting observations, but the authors have not adequately integrated their results with the existing literature. I have concerns about the conclusions and interpretations.

1) An anterior rotational movement characterized by [27], is largely overlooked, but has significant implications for these analyses: what the authors refer to as the optic vesicle temporal domain (where fsta is expressed) gives rise to the dorsal domain (180° away from the optic fissure). Yet the authors describe the dorsal region as having notably less flow. Is this the prospective nasal region? Clarification, and placing these results in the context of previously characterized movements would be useful.

2) The authors show BRE reporter expression in the dorsal domain at optic cup stage. At least one other BMP inhibitor, bambi, is already known to be expressed in the prospective dorsal domain (first temporal, then dorsal) throughout optic cup development (11). The coincidence of BMP inhibitor expression and BRE reporter expression suggests a potential negative feedback loop, though BMP signaling is not completely abolished in the dorsal domain, based on phospho-Smad staining (11) and BRE reporter expression. Therefore, fsta expression (and expression of other BMP inhibitors) may not mark a zone of repressed BMP signaling, as the authors have proposed.

3) The authors argue that BMP overexpression driven by rx2 does not lead to fate changes in the eye, by demonstrating unchanged expression of vax2. However, the embryo appears to be ∼18 somite stage (this is a guess given the lack of staging)? This reviewer was under the impression that the rx2 promoter does not begin to induce transcription until ∼14 somite stage, therefore, it is unclear if vax2 expression would be altered in such a short time. A later in situ (prim–5) would be a better control, especially given strong induction of the BRE and later phenotypes. Any clarification on promoter activity and timing would be welcome.

4) The authors argue that there is no BMP activity in the early optic vesicle by using the BRE reporter line. Is it clear that this line reports all BMP activity? There is evidence that BMP signaling is active in the early optic vesicle: [11], demonstrate strong phospho-Smad staining in the optic vesicle temporal region (which gives rise to the dorsal domain) at 6 and 10 somite stages.

5) Cell movement from lens-averted to lens-facing domain has previously been analyzed via live imaging (27), cell tracking and volume measurements (20). A novel finding would be when such flow ceases, thereby defining the definitive stem cell compartment; this was never demonstrated previously. The authors' movies are likely to contain that information, as the last embryo in Figure 2 appears older than prim–5, based on lens morphology.

6) Embryo ages (somite stages or hours post fertilization) are lacking throughout the manuscript, making it difficult to place these movements in context of other known events.

---

## [Author Response]

While all three reviewers find the study interesting, there are some substantive reservations over the manuscript in its current form. A major criticism is the lack of citation and acknowledgement of previous literature. A number of papers have been suggested by the reviewers, all of which should be considered, cited and discussed carefully. In addition, the conclusions and interpretation of the findings should be adjusted to take full account of these earlier studies.

In addition, some further experimental work should be done to strengthen the manuscript. A fuller description, at multiple embryonic stages, of the BRE:GFP expression domains in the eye, and correlation with expression of both P–Smads and BMP inhibitors (fsta, bambi), is required. A later stage in situ showing vax2 expression following BMP overexpression should also be shown. If additional data (such as the origin of the stem cell compartment, or tracing the patch of lens-averted epithelium that forms the RPE) can be extracted from the existing movies (or generated with new ones), this information would be extremely valuable and would enhance the significance and novelty of the findings.

A number of other suggestions have been suggested by the reviewers to improve the manuscript, including adding more detail to the Experimental Procedures section and clarifying the data provided in the Table. It will be essential to provide accurate embryo staging and labeling throughout.

The full reviews are appended below.

Reviewer #1:

This is an interesting report documenting the cell movements that contribute to formation of the zebrafish neuroretina. The authors demonstrate that much of the retina derives from lens-averted epithelium, which moves round and into the lens-adjacent layer in directed movements of the cell sheet akin to gastrulation. These movements continue in the absence of cell proliferation, and account for the apparently paradoxical increase in surface area of the neuroretina, despite the decrease in apical surface area of individual cells in this tissue. It is proposed that a small patch of the lens-averted epithelium gives rise to the retinal pigmented epithelium (RPE).

Some of the imaging is spectacular and supports the conclusions of the paper well. However, there are also a number of deficits that should be addressed. These are listed below.

General points:

The authors place great importance on the retinal stem cell niche, but have no markers to show this population. Can they be visualised in some way?

This is a very interesting aspect that we have elaborated further in the revised manuscript. The retina specific homeobox transcriptions factor Rx2 in fact represents a marker for retinal stem cells at the relevant stages of the manuscript (see attached manuscript by Reinhardt, Centanin et al., submitted elsewhere). It is initially expressed in all cells of the evaginated optic vesicle (likely stem cells) and as development proceeds is limited to the CMZ. At later stages (once the retinal cell types enter terminal differentiation after the completion of optic cup formation) its expression is additionally found in Müller Glia cells and photoreceptors. The stem cells of the neuroretina and the Retinal Pigmented Epithelium (RPE) are located within the CMZ. Using an inducible CRE based lineage analysis we show in the accompanying manuscript that Rx2 cells of the CMZ represent stem cells. An individual Rx2 positive cell is multipotent and gives rise to all retinal cell types (Reinhardt, Centanin et al., submitted). This is now explicitly stated in the revised version of the manuscript where we refer to rx2 as retinal stem cell marker. The submitted manuscript of Reinhardt, Centanin et al. has been attached.

They also place importance on the role of the developing RPE in driving the cell movements, but the cell movements of this layer are not described well at all, and the small patch of cells that is proposed to become the RPE is not identified or followed in any of the figures or movies. It will be important to show this to confirm the assertion that only a small patch of the lens-averted epithelium develops into the RPE.

This is an important point, and we thank the referee for this comment. In the revised version of the manuscript we now put emphasis on the developing RPE. We use two features to identify early RPE: The absence of Rx2 expression as well as the specific flat morphology of the RPE cells (and nuclei). Using those criteria we follow the edge of the RPE, best visualized in Video 3. We show the developing RPE in between the arrows, which mark the border of RPE. In addition, Figure 1 is highlighting the change in cell shape the RPE precursors undergo, concomitant with their loss of Rx2 expression (between asterisks in Figure 1).

We have taken care to clearly state these important points in the revised manuscript.

Numbers and quantitation: more detail is needed throughout, e.g. for the phenotypes of 'variable expressivity' (eleventh paragraph) resulting from pan-ocular expression of BMP4.

The referee is right. Our statements were misleading and the variability was due to genetic variability in the analyzed lines. We had initially mentioned the variable expressivity to indicate a graded activity of BMP4, which however was apparently not fully explained.

In order to avoid that confusion we now only focus on one phenotypic class and have removed the misleading term from the revised manuscript.

Figures:

Scale bars are needed on the figures, especially on Figure 1, where the surface area is estimated from linear dimensions drawn onto sections. The scale is needed to ensure the data shown in the figure panels match up with the values listed in the Table.

Scalebars have been added to the figures, which were used for quantitation.

Stages/hours post fertilisation should also be listed in the legend or directly on the panels for all figures.

Information about the developmental stages has been added to the figure legends.

Figure 1–figure supplement 1 should show a control (H2BGFP without aphidicolin). If this is provided by Figure 2, there should be a reference to this figure in the legend to Figure 1–figure supplement 1, so that a comparison can be made.

The new figure (Figure 2—figure supplement 1) presenting this data is connected to Figure 2, which is showing the control data. As requested we have put a reference to the corresponding figure legend.

Are the small bright spots the dying cells? How many embryos were treated, and with what concentration of aphidicolin? This information does not appear to be provided anywhere. How was the block in cell proliferation confirmed? Was this by counting nuclei, or lack of mitotic figures?

The referee is right, we also believe that the small bright spots represent condensed nuclei of apoptotic cells. Notably this is not affecting the described neuroretinal flow. We added the necessary information regarding the aphidicolin treatment in the revised manuscript.

Briefly, we pretreated for 5 hours before imaging with a concentration of 10µg/ml (Serva, cat:13696) 12 embryos. One of those was subjected to in vivo time-lapse analysis.

The efficacy of the treatment was addressed by analyzing nuclei in mitosis (positive for the expression of phospho-histone H3. At 21.5 hpf pHH3 positive nuclei were counted in central sections of four control (average: 21) and experimental (average: 6) retinae respectively. We have now included the anti-pHH3 immunohistochemical stainings to the revised manuscript (Figure 2—figure supplement 1) showing the aphidicolin mediated inhibition of cell proliferation.

Figure 4–figure supplement 1A, panel 2: This picture is not of publication quality, it is completely out of focus, with no morphological detail visible.

Figure 4–figure supplement 2: It would be helpful to show here a stage series, showing when the BMP reporter normally begins to be expressed in the eye, and how levels of expression compare with other areas of BMP activity in the embryo. Ideally, an additional confirmation of BMP activity, e.g. by P-Smad levels, should be shown; alternatively, more information about this transgenic line or a citation is needed in the Experimental Procedures.

The referee is right and we have restructured that entire part also in response to the general request and the comments of the other referees. Figure 4–figure supplement 1A has been removed. Instead we show a new Figure 4 which represents the dynamics of fsta and bambi expression along with indicators for BMP activity (anti-pSmad 1/5/8 immunohistochemical stainings and BRE::GFP expression) at relevant stages for optic cup formation. The BRE:GFP zebrafish have been provided by Beth Roman. The original paper (21) has (and had) been cited. The combination of BMP signaling modulators fsta and bambi with indicators for BMP activity at the key stages of the vesicle to cup transition now provides all necessary evidence at a glance. The text and figure legends have been revised accordingly.

Figure 5–figure supplement 1 merge: The box drawn in the first panel does not appear to correspond to the area shown in the enlargements.

We have taken care that the box moved perfectly corresponds to the close up, thanks for pointing this out.

Figure 5–figure supplement 4 needs repeating and improving. A control is needed to show expression levels in non-transgenic embryos. A higher power picture of the staining in the eye would be useful; the outline and arrows currently overlie the staining, making it difficult to see.

We followed the referee’s advice and repeated the entire series of in situs of vax2 on wild type and rx2::BMP4 embryos. Incorporating the suggestions of the other referees we now show embryos at a later stage (28 hpf) to address the potential impact of ectopic BMP4 on ventral patterning of the retina (Figure 6—figure supplement 4). We show the eyes of control and rx2::BMP4 embryos from different angles (lateral and ventral). The expression of vax2 in the ventral part of the retina persists even under conditions of BMP expression in the entire optic vesicle. This indicates that the observed morphological alterations observed under those conditions are not due to a miss-patterning of the ventral retina but can rather be attributed to the antimotogenic effect of BMP4. We have taken care to place arrows and outline outside of the staining.

Experimental Procedures: More detail is needed for description of the transgenic lines. Are these already published? In which case, a citation is needed. If not published, more description of the constructs used is needed. A ZFIN reference should be given for each line, if available.

This is an important point. All published transgenic lines were properly referenced in the submitted version and are as well in the revised version. All of the components of the lines generated in the context of this manuscript are referenced in detail in the experimental procedures section of the revised manuscript. The details are found in the revised experimental procedures.

Immunohistochemistry: Not enough detail is given here for others to be able to replicate the experiments. What do the acronyms stand for, what dilutions of primary and secondary antibodies were used, and what concentrations of DAPI, DMSO, Triton, etc.?

We have taken care that the information now presented allows full reproduction of the experiments. The relevant information is provided in the experimental procedures section of the revised manuscript.

Table:

The Table is difficult to understand. What are the numbers? Do they refer to measurements from a single embryo, or are they mean values from multiple embryos? (In which case, SEM or SD should be given). If from a single embryo, it is difficult to draw any firm conclusions from the data. In the text, phase 2 is described as a fast, smooth flow, but in all cases the effective speed shown during phase 2 in the table appears to be slower than the values shown for phase 1. The total distance moved in phase 2 needs to correlate with the distances shown on the figures, which is why scale bars on the figures are needed. It would be helpful to have a full stop instead of a comma for the decimal point, as 10,000 could be read as 'ten thousand'.

The referee is right, the table is difficult to understand. It reflects the attempt to provide quantitation of the movies to allow a more intuitive perception of the situation. Apparently that attempt failed. Since the table does not provide any information that is not also provided by the movies and it appears more confusing than enlightening, we have decided to remove it from the manuscript to enhance the clarity of our arguments. We thank the referee for pointing out this problem.

Movies:

The movies are very helpful, but more information is needed in the legends that describe these. What are the times over which they were taken, what was the frame rate, and what stage are the embryos? As for the figures, this information is needed to correlate with the values shown in the table, and should be given in the movie legends. Video 8 is of lower quality and resolution than the others, and should be improved.

We now provide the information requested for each of the movie files in the revised version of the manuscript. Video 8 (Video 10 in the revised manuscript) represents imaging deep inside the optic cup. It has been taken under the same imaging settings as all the other movies. Due to scattering of the overlying tissue the quality could only be partially improved. This improved movie is now provided as revised Video 10.

Reviewer #2:

This manuscript analyses the cell dynamics accompanying optic cup formation in the zebrafish and shows that cells in the “lens-averted” region of the optic vesicle are incorporated into the “lens-facing” region. It further shows that the inhibition of BMP is required for this process to occur efficiently. The study has interesting implications to understand the origin of optic cup domains. In addition, it provides a new interpretation for phenotypes that up to now had been interpreted as transdifferentiation of RPE into retina, and highlight the power of live imaging to provide an accurate interpretation of phenotypic outcomes. I consider the work a valuable contribution to our current knowledge on eye morphogenesis and patterning. However, I have important concerns with the way the significance of previous works analyzing this same process has been minimised by the authors, and with the interpretation of some of their observations.

The finding of optic vesicle cells ingressing into the future retinal layer from the prospective RPE layer is not new. It has been extensively described by Picker et al., PLoS Biology, 2009 and Kwan et al., Development, 2012. I disagree with the way the authors seem to remove significance from those papers, and I consider they should be appropriately cited. In particular, Picker et al. clearly and extensively showed that prior to optic cup formation, the optic vesicle is mostly formed by future retinal cells, and that the lens-averted layer of the primordium constitutes the origin of the temporal retinal domain.

We fully agree with the referee that Picker et al., as well as Kwan et al., described flow over the distal rim of the developing optic cup and are sorry for not adequately pointing that out in the initial version. It was never our intention not to acknowledge these findings or to remove significance from them.

In the revised version of the manuscript we substantially extended this paragraph to give full credit to previous research. We added the reference to a paper published before Picker et al. and Kwan et al. speculating about such a flow (22). Importantly the flow in the temporal domain was then shown by Picker et al. and confirmed by Kwan et al. The nasal flow was not demonstrated and the connection of flow to the origin of the optic fissure had not been addressed before. The aspect of the dorsal pole, showing no flow, had also not been noted before.

This does not mean that I do not consider the current work valuable. The authors do provide a detailed analysis of the consequences this morphogenetic process has to understand the origin of different optic cup domains (in particular the CMZ and the RPE), and dissect part of the molecular mechanisms controlling it.

We thank the referee for this supportive statement and hope that our changes have clarified the issue.

The authors state that follistatin is expressed in the temporal domain, yet their in-situ (Figure 4) clearly shows expression in the prospective dorsal portion of the optic vesicle. At early stages, the optic vesicle in zebrafish has not acquired its final disposition and temporal is located along the ventral portion of the optic vesicle. Follistatin expression is stronger at the distal-most region of the optic vesicle, both dorsal and ventral, which corresponds to prospective dorsal (see Veien et al., Development, 2008). Indeed, this would be more consistent with previous reports showing a requirement of differential BMP signalling along the dorso-ventral, and not the naso-temporal, axis.

We agree with the referee that this is an important issue. Since the orientation of the vesicle and the localization of BMP activity with respect to the expression of BMP modulators was also brought up by the other referees we now provide a new Figure 4 for further clarification. We added more data showing the expression of fsta, bambi, as well as phosopho SMAD and the active BMP sensor at different developmental stages (please see revised Figure 4).

The point of eye rotation, which has been described in zebrafish does not pose a problem for the relative annotation of expression patterns during eye cup formation.

In a manuscript of Schmitt and Dowling (1994) two important phases of “eye rotation” have been described, a slight early rotation (10-12SS) and a more intense rotation at a later phase (24-36 hpf). Our analysis clearly shows that the neuroretinal flow described in our manuscript starts after the first, slight rotation and has already ceased before onset of the intense rotation of the optic cup (24 to 36 hpf, Schmitt and Dowling, 1994). We describe the formation of the optic fissure by retinal flow in between the two phases of eye rotations. This is in line with the timing of development described by Schmitt and Dowling (1994).

In the paper by Veien et al., the analyses started at an earlier developmental stage than our analyses likely addressing the first phase of “eye rotation”. As described above, the neuroretinal flow described in our manuscript occurs in between the rotational stages. Thus the relative positions remain stable in the phase of eye formation we are discussing here. Therefore all the patterns described are not apparently affected by the known phases of eye rotation. Our analysis furthermore indicates that expression patterns are dynamic with respect to transcriptional control and are therefore of limited value for linking fates and position (compare the expression of fsta and bambi in Figure 4).

With respect to the impact of the modulation of BMP signaling in the nasal and temporal domain the new Figure 4 now provides compelling expression data that fully support the hypothesis presented in the manuscript (please see the revised Figure 4).

In the same line with my first comment, I find the authors should be more rigorous with their citation of previous work. In the first sentence of their manuscript and based on a previous report from the authors from 2006, they state that optic vesicle evagination occurs by single cell migration. However, since then there have been additional papers analyzing dynamically this process and showing results that support alternative explanations for the active behaviours of eye field cells. These papers (England et al., Development, 2006 and Ivanovitch et al., Dev Cell, 2013) should be acknowledged, and the first statement softened.

It was not our intention not to acknowledge previous work. Although this aspect is only part of our Introduction, we extended this paragraph and cited the mentioned papers. We have tried to present a synthesis of the not so different perspectives promoted by the respective authors.

Reviewer #3:

Given the involvement of BMP signaling in ocular patterning, it is exciting to find that overexpression of BMP can affect specific cell movements. The manuscript contains some interesting observations, but the authors have not adequately integrated their results with the existing literature. I have concerns about the conclusions and interpretations.

1) An anterior rotational movement characterized by [27], is largely overlooked, but has significant implications for these analyses: what the authors refer to as the optic vesicle temporal domain (where fsta is expressed) gives rise to the dorsal domain (180° away from the optic fissure). Yet the authors describe the dorsal region as having notably less flow. Is this the prospective nasal region? Clarification, and placing these results in the context of previously characterized movements would be useful.

We thank the referee for this comment. We realized that this issue needs further clarification. In the paper of Schmitt and Dowling (1994) two important phases of “eye rotation” have been described, one slight rotation (10-12SS) and a more intense rotation at a later phase (24-36 hpf).

Our analysis clearly show that the neuroretinal flow, which we describe, starts after the first, slight rotation and has already ceased before onset of the intense rotation of the optic cup (24 to 36 hpf, Schmitt and Dowling, 1994). Further, based on our 4D data we describe the formation of the fissure by the retinal flow, in between the two phases of eye rotations. This is in line with the timing of development described by Schmitt and Dowling (1994). However, our 4D data clearly show a different developmental process behind the formation of the optic fissure.

Importantly Picker et al. started their analyses at 10SS, therefore including at least the first phase of “eye rotation”. It is also important to consider the axis of the developing eye in relation to the axis of the body. Especially the heavy rotation of the optic cup aligns the axis of the eye and the axis of the body at this late developmental stage. With respect to the onset of neuroretinal flow, the dorsal and ventral poles relate to the optic vesicle and later define also the axis of the eye. However, as described above, the axis of the eye is only later aligned to the axis of the zebrafish body.

Based on our 4D analysis of the eye it is apparent that gene expression patterns are of limited value as cell/tissue fate markers (please compare the expression patterns of bambia to fsta in the revised Figure 4).

For addressing the neuroretinal flow by the expression of the BMP antagonists, their expression pattern at the optic vesicle stage are of importance. Here especially the expression pattern of fsta can explain the modulation of BMP signaling in the nasal and temporal domain facilitating the neuroretinal flow (please see the revised Figure 4).

2) The authors show BRE reporter expression in the dorsal domain at optic cup stage. At least one other BMP inhibitor, bambi, is already known to be expressed in the prospective dorsal domain (first temporal, then dorsal) throughout optic cup development (11). The coincidence of BMP inhibitor expression and BRE reporter expression suggests a potential negative feedback loop, though BMP signaling is not completely abolished in the dorsal domain, based on phospho-Smad staining (11) and BRE reporter expression. Therefore, fsta expression (and expression of other BMP inhibitors) may not mark a zone of repressed BMP signaling, as the authors have proposed.

We are thankful for this comment. We included the BMP antagonist bambi into our analyses and noted the strikingly overlapping patterns of BMP signaling activity and bambi expression (revised Figure 4). The referee is right in stating that bambi is not totally antagonizing BMP signaling but rather is modulating it or may be even modulated by it. With bambi alone it would be hard to explain the modulation of BMP signaling to facilitate the neuroretinal flow. However, fsta shows a different expression pattern (please see the revised Figure 4), able to explain the facilitation of the flow nicely.

3) The authors argue that BMP overexpression driven by rx2 does not lead to fate changes in the eye, by demonstrating unchanged expression of vax2. However, the embryo appears to be ∼18 somite stage (this is a guess given the lack of staging)? This reviewer was under the impression that the rx2 promoter does not begin to induce transcription until ∼14 somite stage, therefore, it is unclear if vax2 expression would be altered in such a short time. A later in situ (prim–5) would be a better control, especially given strong induction of the BRE and later phenotypes. Any clarification on promoter activity and timing would be welcome.

In situ hybridizations with a vax2 probe have been added (Figure 6—figure supplement 4). This figure shows the staining of control vs. rx2::BMP4 embryos at a later developmental stage (28 hpf). Although the vax2 domain is smaller in the rx2::BMP4 embryos, compared to the controls, it is not absent. Hence, the reason for the arrested flow (16.5–21.5) is unlikely an early mis-differentiation of the ventral domain.

4) The authors argue that there is no BMP activity in the early optic vesicle by using the BRE reporter line. Is it clear that this line reports all BMP activity? There is evidence that BMP signaling is active in the early optic vesicle: [11], demonstrate strong phospho-Smad staining in the optic vesicle temporal region (which gives rise to the dorsal domain) at 6 and 10 somite stages.

We agree with the referee and included next to the analysis of BRE::GFP anti-pSmad 1/5/8 immunohistochemical stainings, both at different developmental stages (Figure 4). Notably the BMP signaling reporter shows a delay of activity, which can be explained by its nature as a transcriptional reporter. This suggests that BMP signaling still is active. This is in favour of a modulative action of the BMP antagonists rather than a total block. We thank the referees for this comment.

5) Cell movement from lens-averted to lens-facing domain has previously been analyzed via live imaging (27), cell tracking and volume measurements (20). A novel finding would be when such flow ceases, thereby defining the definitive stem cell compartment; this was never demonstrated previously. The authors' movies are likely to contain that information, as the last embryo in Figure 2 appears older than prim–5, based on lens morphology.

We agree with the referee that Picker et al., as well as Kwan et al., described flow over the distal rim of the developing optic cup. It was not our intention not to acknowledge these findings. We extended the paragraph discussing this point in the revised version of the manuscript. In addition we added a reference to an additional manuscript speculating about such a flow (22) even before Picker and Kwan.

The flow in the temporal domain was then shown by Picker et al. and confirmed by Kwan et al. The nasal flow was not demonstrated and the flow with respect to the origin of the optic fissure was not addressed before. The aspect of the dorsal pole, showing no flow, was also neither described nor addressed before.

Regarding the forming stem cell compartment (CMZ), this was part of our analyses. We describe the origin of the cells, which eventually reside in the CMZ domain, by tracking them backwards in 4D (Figure 3, Video 5).

6) Embryo ages (somite stages or hours post fertilization) are lacking throughout the manuscript, making it difficult to place these movements in context of other known events.

The developmental stages of embryos have been added to the figure legends.